# Sensor-Level Wavelet Analysis Reveals EEG Biomarkers of Perceptual Decision-Making

**DOI:** 10.3390/s21072461

**Published:** 2021-04-02

**Authors:** Alexander Kuc, Vadim V. Grubov, Vladimir A. Maksimenko, Natalia Shusharina, Alexander N. Pisarchik, Alexander E. Hramov

**Affiliations:** 1Neuroscience and Cognitive Technology Laboratory, Center for Technologies in Robotics and Mechatronics Components, Innopolis University, Universitetskaya Str. 1, 420500 Innopolis, Russia; kuc1995@mail.ru (A.K.); v.grubov@innopolis.ru (V.V.G.); v.maksimenko@innopolis.ru (V.A.M.); alexander.pisarchik@ctb.upm.es (A.N.P.); 2Institute of Information Technologies, Mathematics and Mechanics, Lobachevsky State University of Nizhny Novgorod, 603950 Nizhny Novgorod, Russia; 3School of Life Sciences, Immanuel Kant Baltic Federal University, 236016 Kaliningrad, Russia; nnshusharina@gmail.com; 4Centre for Biomedical Technology, Universidad Politécnica de Madrid, 28040 Madrid, Spain; 5Saratov State Medical University, Bolshaya Kazachya Str. 112, 410012 Saratov, Russia

**Keywords:** perceptual decision-making, ambiguous stimuli, selective attention, top-down processes, beta-band activity

## Abstract

Perceptual decision-making requires transforming sensory information into decisions. An ambiguity of sensory input affects perceptual decisions inducing specific time-frequency patterns on EEG (electroencephalogram) signals. This paper uses a wavelet-based method to analyze how ambiguity affects EEG features during a perceptual decision-making task. We observe that parietal and temporal beta-band wavelet power monotonically increases throughout the perceptual process. Ambiguity induces high frontal beta-band power at 0.3–0.6 s post-stimulus onset. It may reflect the increasing reliance on the top-down mechanisms to facilitate accumulating decision-relevant sensory features. Finally, this study analyzes the perceptual process using mixed within-trial and within-subject design. First, we found significant percept-related changes in each subject and then test their significance at the group level. Thus, observed beta-band biomarkers are pronounced in single EEG trials and may serve as control commands for brain-computer interface (BCI).

## 1. Introduction

Perceptual decision-making is the process of choosing a course of action based on available sensory information [1]. It is a fundamental mechanism enabling our interaction with the environment. Neuroscientists usually analyze perceptual decision-making to understand how the brain structures interact to distinguish and classify information from the senses [2,3,4]. They also consider how external and internal factors affect our perceptual decisions. The internal factors include the human condition, e.g., motivation, fatigue, and alertness [5]. Furthermore, they rely on personal experience and memory [6]. Thus, subjects better respond to the sensory stimuli if they saw them before and increase responses to the repeatedly presented stimuli [7]. External factors reflect the quality of the sensory evidence. If the decision-relevant features or stimuli have well-pronounced distinctive features, e.g., color, they quickly grab our attention and speed up the final decision (bottom-up attention) [8]. When the sensory information is ambiguous and contains many irrelevant features, we voluntarily shift our attention to the most relevant details (top-down attention) [9].

In their early works, Philiastides and colleagues used ERP to characterize neuronal mechanisms of perceptual decision-making. They identified ERP whose latency depended on the strength of evidence. Thus, they reported that the evidence accumulation process starts at ate early processing stages, and its duration varies as a function of the evidence strength affecting decision speed [10]. In 2007, the same authors used fMRI (functional magnetic resonance imaging) and reported that perceptual decision-making includes multiple spatially and temporally dissociated stages. These include early visual perception (early component), task/decision difficulty (difficulty component), and post-sensory/decision-related events (late component) [11]. In 2009, Ratcliff et al. analyzed single trials during a perceptual decision-making task. They found that the late but not the early ERP component correlated with the single-trial behavioral performance. The authors concluded that the information used to drive the decision process—that is, reflected in the late EEG (electroencephalogram) component—is not the same as that which produces the early EEG response. Thus, they distinguished between neural responses to early perceptual encoding and, later, to post-sensory processing that provides the decision-relevant evidence [12]. Note that the authors reported trial-to-trial variability in the decision performance even for identical images. It may reflect the reliance of the evidence accumulation on human internal factors that varied through experiment. One possible mechanism modulating the human state and affecting perceptual decisions is attention. With this in mind, Nunes and colleagues evaluated attention in single trials. Their results suggested that fluctuations in attention to visual stimuli account for some of the trial-to-trial variability in the brain’s speed of evidence accumulation on each trial [13]. Recent work by Herding et al. tested the effect of another human factor on the evidence accumulation process. They reported that subjective assessment of the evidence plays a crucial role in the accumulation process. According to their results, sensory input appeared to be first modulated by the subjectively perceived features and later by its absolute value (i.e., the absolute strength of evidence) [14]. Together, these works suggest that effects of attention and subjectiveness enhance when the evidence quality is poor, e.g., noisy or ambiguous. When the sensory information is not sufficient to reliably disambiguate the nature of a stimulus, it requires more attentional demands and efforts to acquire decision-related features. At the behavioral level, the ambiguity of sensory information influences many oculomotor variables, including fixation frequency and duration, the frequency, peak velocity, and amplitude of saccades, and phasic pupil diameter [15]. At the neural activity level, increasing ambiguity induces high β-band power across the fronto-parietal areas, reflecting attentional modulation and the involvement of the internal processes e.g., memory and subjective experience [6,14].

Studying these attentional components also has a practical meaning. Attention is a fundamental cognitive function that develops in childhood and declines in old age. Thus, monitoring this vital brain function enables the brain-computer interface (BCI) development for the prediction and prevention of attentional deficits and disorders [16,17].

Attentional mechanisms are actively studied using within-subjects design and require large groups of participants. In contrast, BCI studies use within-trial methods due to the inter-subject variability [18]. A common way of training BCI systems is subjecting a pre-trained algorithm to the additional calibration session, optimizing its performance for a particular subject. On the one hand, a pre-trained algorithm will be effective if it utilizes the between-subject features. On the other hand, calibration requires subject-specific features to learn. Thus, revealing potential electrophysiological biomarkers for BCI may require mixed within-trial and within-subject analysis.

A bulk of literature suggests that attention modulates neuronal activity in the α- and β-frequency bands (see [4] for the literature review). On one hand, activity in these bands reflects the degree of attention in general regardless of the task [19]. Thus, measuring the prestimulus α- and β-band power enables predicting the performance of the ongoing stimuli processing. Different studies report that low α- and high β-band power reflect high attention therefore, predicting high performance and vice-versa [20]. On the other hand, some components of attention influence perceptual decisions when the sensory information is presented. Thus, stimulus onset involuntary draws our bottom-up attention. When the stimulus is ambiguous, its processing mobilizes top-down attentional components e.g., selective attention to facilitate the evidence accumulation process [21]. Based on these studies, we suppose that stimulus-related attentional components also modulate the α- and β- band activity on the interval following stimulus onset depending on the attentional demands, e.g., the strength of evidence. Following [13], we suppose that measuring power in these bands enables evaluating attentional demands on the single trials providing potential for BCI applications.

To test this hypothesis, we considered the α- and β-band wavelet power of an EEG signal registered during the perceptual decision-making task. This task required participants to respond to the particular orientation of the ambiguous bistable stimuli [22,23,24]. According to the previous work, we supposed that high ambiguity makes subjects rely on the top-down rather than bottom-up mechanisms [25]. Thus, we compared the stimuli with low and high ambiguity to reveal biomarkers of the bottom-up processes. First, we used the within-trial design to quantify the perceptual process for each subject. Then, we contrasted obtained quantities between low and high ambiguities following within-subject statistics. We observed that the α-band power decreased after the stimulus onset over most sensors. In contrast, the β-band power increased after the stimulus onset. The high ambiguity induced high frontal β-band power at 0.3–0.6 s post-stimulus onset. We suppose that ambiguous stimuli may induce the frontal β-band power evidencing internal top-down processes that control evidence accumulation to disambiguate stimulus, interpret it, and make an appropriate decision.

Since the observed β-band biomarkers are significant in within-trial analysis, they may be pronounced in single EEG trials and may serve as control commands for BCI. Moreover, their significance in the group of subjects suggests that subjects share a common perceptual mechanism. Thus, revealed biomarkers may increase the effectiveness of BCI calibration for a particular user and its resistance to the between-subject variability.

## 2. Materials and Methods

### 2.1. Participants

Unpaid volunteers participated in the experiment. All subjects were conditionally healthy with normal acuity (thus, wearing no glasses or contact lenses) and with no history of eye diseases or pathologies. The experimental group consisted of 20 subjects (16 males and 4 females) at the age of 20–36. The volunteers were informed about the goals and methods of the experiment and possible inconveniences related to the experimental procedure. They were able to ask any questions about the experiment and received satisfactory answers. All subjects filled and signed a written consent form before their participation. The experimental study was designed and carried out in accordance with the Declaration of Helsinki. All works were approved by the local research ethics committee of the Innopolis University.

### 2.2. Visual Stimuli

We used Necker cube images as the visual stimuli [26,27]. The Necker cube is a drawing of a 3D-cube with transparent faces and visible edges projected on a 2D surface. Due to the specific arrangement of the edges, an observer with normal perception interprets the Necker cube as a 3D-object. The main feature of the Necker cube as a visual stimulus is the ambiguity of its perception: The cube can be interpreted as “left-oriented” or “right-oriented”. This bistability of perception can be further amplified and controlled by changing the contrast between some of the internal edges. In our work, we varied the contrast of three edges in the lower-left middle part of the image and three similar edges in the upper-right middle part (see Figure 1A).

First, we introduced parameter I={0.15,0.25,0.4,0.45,0.55,0.6,0.75,0.85} defining the inner edges contrast. It reflects the intensity of three lower-left lines, while 1−I corresponds to the intensity of three upper-right lines. The parameter *I* can be defined as I=1−y/255, where *y* is the brightness level of three lower-left lines using the 8-bit gray-scale palette. The value of *y* varies from 0 (black) to 255 (white). The Necker cube images with the different contrast parameters are shown in Figure 1A. Then we introduced stimulus ambiguity, *a* in the following way. We supposed that for I=0 stimulus is unambiguously left-oriented, while for I=0.5 its features barely reflect the orientation. Varying *I* from 0 to 0.5, we increase the ambiguity of the left-oriented cube making it totally ambiguous at I=0.5. Thus setting a=0% ambiguity for I=0 and a=100% ambiguity for I=0.5, we suggest that stimuli with I={0.15,0.25,0.4,0.45} correspond to a={30%,50%,80%,90%} ambiguity. Similarly, for the right-oriented stimuli, we obtain that cubes with I={0.85,0.75,0.6,0.55} also correspond to a={30%,50%,80%,90%} ambiguity. Finally, to exclude effects of the stimulus orientation (including the effects associated with the formation of the motor response), we combined left- and right-oriented stimuli for each ambiguity.

### 2.3. Experimental Protocol

During the experiment the subject was sitting in the chair with a monitor screen before their eyes and a joystick with two buttons, left and right, in their hands. The experiment lasted for ∼40 min and included an active session and short (∼120 s) background activity recording before and after the active session. During the active session, visual stimuli were demonstrated to the subject. Each visual stimulus consisted in demonstration of one Necker cube. The cube was demonstrated on the computer screen with a white background and red dot in the center to attract the subject’s attention to the cube. We used a 24”-LCD monitor with a spatial resolution of 1920×1080 pixels and a 60 Hz refresh rate for stimulus presentation. Eye-to-monitor distance was 70–80 cm with visual angle of ∼0.25 rad. The Necker cube size on the screen was 14.2×14.2 cm.

Each of the 8 cubes from the set were presented 50 times, which lead to total 400 presentations. The structure of each presentation is illustrated by Figure 1B. Each *i*-th presentation consisted of the cube’s presentation ($\tau_{i}$) and pause($\gamma_{i}$) before the next, (i+1)-th, trial.

The subjects were instructed to look at the cube and interpret it as either left- or right-oriented and then to press the corresponding button on the joystick—the left button with the left thumb and the right button with the right thumb.

In case of consecutive presentations previously demonstrated bistable images could affect the perception of subsequent images, known as the “memory effect” [28]. For example, an observation of several left-oriented cubes in a row could stabilize the subject’s perception so the next right-oriented cube can be falsely interpreted as left-oriented. To reduce the influence of this effect we added a variation in the experimental design: The type of the Necker cube (the control parameter *a*) was chosen randomly in each trial and the length of each cube’s presentation varied in range τ∈[1.0,1.5] s. Additionally, during the pause between consecutive cubes an abstract image was demonstrated (see Figure 1B) to draw the observer’s attention and to make a perception of each Necker cube independent from the previous ones. Length of pause varied in range γ∈[3.0,5.0] s.

### 2.4. EEG Acquisition and Preprocessing

To record EEG signals we used an electroencephalograph “Encephalan-EEG-19/26” (Medicom MTD company, Taganrog, Russian Federation) with a two-button input device (joystick). EEG data were acquired using a monopolar registration method with the following set of electrodes: 31 recording EEG electrodes placed according to an extended “10–10” scheme, two reference electrodes (A1 and A2) placed on earlobes, and a ground electrode (N) placed above the forehead. EEG signals were recorded with the cup adhesive Ag/AgCl electrodes. During the experiments electrode, impedances were monitored to ensure the high quality of recorded EEG data. Common impedance values varied in a range of 1–25 kΩ.

To remove low- and high-frequency noises as well as 50-Hz interference from the power grid, raw EEG data were filtered by the band-pass filter (cut-off frequencies 1 and 70 Hz) and 50-Hz notch filter. Recorded EEG signals were divided into a number of trials. Each trial was associated with a single Necker cube demonstration and included 0.5 s before the cube’s presentation and 1.5 s after the presentation, thus, forming a 2-s interval.

To reduce the presence of physiological artifacts on EEG signals, we performed preprocessing. To remove eye-movement and cardiac rhythm artifacts, we implemented a method based on empirical mode decomposition [29]. Artifacts were considered removed if their amplitude reduced after preprocessing to at least 30% of the initial value. However, some artifacts could not be properly removed, so trials with such artifacts were rejected. This led to reducing the overall number of trials for each subject from 400 to ∼350. Additionally, muscle artifacts are known to be very difficult to remove on the preprocessing stage, so we asked participants to take a comfortable pose during the experiment to minimize spine, neck, and shoulder muscle tension.

To analyze the spatial properties of multichannel EEG data, we divided EEG channels on an extended “10–10” scheme into 7 regions (Fp, F, FC, C, CP, P, and O) as shown with the shaded bands in Figure 1C. These regions are chosen to include channels that belong to one longitudinal brain axis and both hemispheres. Lateral effects were not considered since each group of visual stimuli (LA and HA) incorporates left- and right-oriented Necker cubes in equal ratio.

### 2.5. Behavioral Estimates

After the EEG preprocessing procedure, some trials were excluded due to high-amplitude artifacts. To keep the number of EEG trials constant for each ambiguity, we consider 320 trials out of the initial 400 including 40 trials for each ambiguity. For each stimulus, we estimated a behavioral response by measuring the response time (RT) corresponding to time passed from stimulus presentation to button pressing (Figure 1C).

Figure 2A demonstrates the typical RT distributions obtained for each ambiguity for one single subject. One can see that the median RT shifts to higher values when the ambiguity increases. Simultaneously, an increased median RT accompanies increased dispersion leading to the overlapping of the distributions. To minimize overlap for each ambiguity, we consider 20 trials for which RT lies between the 25th and 75th of the distribution.

Figure 2B demonstrates that the median RT for the group of participants (group mean ± SE) increases with increasing stimulus ambiguity (* p<0.05, *t*-test with Bonferroni correction).

In this work, we reduce the number of experimental conditions considering a={30%,50%} as the low-ambiguity (LA) stimuli and a={80%,90%} as high-ambiguity (HA) stimuli. Each group included 80 stimuli (20 per each ambiguity, 40 per each orientation). The reaction time (group mean) was 0.84 s for LA and 1.09 s for HA stimuli (Figure 2C). This simplification is based on our previous works on the Necker cube images [7,25]. It enables revealing effects of ambiguity and provides a sufficient number of trials to minimize additional effects of orientation, a bias of the presentation moment, and the previously presented stimulus [7]. Simultaneously, according to Brunye et al., very different perceptual and cognitive processes may underlie decision making at low, medium, and high uncertainty levels [15]. Thus, our future studies will test this effect in Necker cubes with low, moderate, and high ambiguity.

Many studies used ambiguous visual stimuli, including Necker cube to analyze spontaneous perceptual reversals. They involve the presentation of totally ambiguous stimulus whose interpretation spontaneously alternates under endogenous or exogenous factors [30,31]. Our paradigm excluded completely ambiguous stimuli. Manipulating with the internal edge contrast, we vary ambiguity in the range where each stimulus had a particular interpretation defined by its morphology. The subjects were able to report a correct stimulus interpretation above chance. Their mean error rate (Figure 2D) was 8.95% for HA stimuli and 1.65% for LA stimuli (Z=3.5,p<0.001, Wilcoxon test).

### 2.6. Signal Analysis

We analyzed EEG spectral power in the α- and β-frequency bands, using continuous wavelet transform (CWT) [32]. First, the wavelet power spectrum $E^{n}\left(f,t\right)={\left(W^{n}\left(f,t\right)\right)}^{2}$ was calculated for each EEG channel $X_{n}\left(t\right)$ in the frequency range f∈[1,30] Hz including both α and β ranges. Here, $W^{n}\left(f,t\right)$ is the complex-valued wavelet coefficients calculated as:(1)$$W^{n}\left(f,t\right)=\sqrt{f}\int_{t-4/f}^{t+4/f}X_{n}\left(t\right)\psi *\left(f,t\right)dt,$$
where n=1,…,N is the EEG chanel number (N=31 is the total number of chanels used for the analysis) and “*” defines the complex conjugation. The mother wavelet function ψ(f,t) is the Morlet wavelet which is defined as:(2)$$\psi \left(f,t\right)=\sqrt{f}\pi^{1/4}e^{j\omega_{0}f\left(t-t_{0}\right)}e^{f{\left(t-t_{0}\right)}^{2}/2},$$
where $\omega_{0}=2\pi$ is the wavelet parameter. According to [33], the Morlet wavelet has advantages in the analysis of EEG recordings that includes many rhythms and oscillatory patterns. The comparison of EEG analysis studies with different mother wavelets [32] shows that the Morlet wavelet provides a clear wavelet surface and better overall resolution in the time-frequency domain.

For α- and β-frequency bands, the wavelet amplitudes $E_{\alpha }^{n}\left(t\right)$ and $E_{\beta }^{n}\left(t\right)$ were calculated as:(3)$$E_{\alpha ,\beta }^{n}\left(t\right)=\frac{1}{\Delta f_{\alpha ,\beta }}\int_{\Delta f_{\alpha ,\beta }}E^{n}\left(f^{\prime },t\right)df^{\prime },$$
where $\Delta f_{\alpha }=8-12$ Hz, $\Delta f_{\beta }=15-30$ Hz. In order to neglect the changes of the overall EEG signal amplitude, the values (Equation 3) were normalized to the EEG spectral amplitude in the 1–30 Hz frequency band.

The time-series of the wavelet power (Equation 3) was calculated for the whole time of the experimental session and then was split into the time segments $\tau_{\mathrm{pre}}^{i}=0.5$ s and $\tau_{\mathrm{post}}^{i}=0.5$ s, before and after the *i*-th visual stimulus presentation (see Figure 3):(4)$${\langle E_{\alpha ,\beta }^{n}\rangle }_{\tau_{\mathrm{pre}}^{i},\tau_{\mathrm{post}}^{i}}=\int_{\tau_{\mathrm{pre}}^{i},\tau_{\mathrm{post}}^{i}}E_{\alpha ,\beta }^{n}\left(t^{\prime }\right)dt^{\prime }.$$

For each stimulus ambiguity, the difference between ${\langle E_{\alpha ,\beta }^{n}\rangle }_{\tau_{\mathrm{pre}}^{i}}$ and ${\langle E_{\alpha ,\beta }^{n}\rangle }_{\tau_{\mathrm{post}}^{i}}$ for the *n*-th EEG sensor was analyzed statistically via the paired samples *t*-test based on 20 trials. Figure 3 demonstrates the estimation of the change between ${\langle E_{\alpha ,\beta }^{n}\rangle }_{\tau_{\mathrm{pre}}^{i}}$ and ${\langle E_{\alpha ,\beta }^{n}\rangle }_{\tau_{\mathrm{post}}^{i}}$ on the basis of an illustrative example. The dependences $E_{\alpha ,\beta }^{n}$ are shown for 20 trials as the median and 25th–75th percentiles (Figure 3A,C). In Figure 3B,D the corresponding values ${\langle E_{\alpha ,\beta }^{n}\rangle }_{\tau_{\mathrm{pre}}^{i}}$ and ${\langle E_{\alpha ,\beta }^{n}\rangle }_{\tau_{\mathrm{post}}^{i}}$ are shown as medians (bars), 25–75 percentiles (box) and outlines (whiskers).

To address the multiple comparison problem (MCP), we used the permutation tests in conjunction with cluster-based correction. A cluster was significant when its *p*-value was below 0.025, corresponding to a false alarm rate of 0.05 in a two-tailed test. The number of permutations was 2000. The sensors that passed a MCP correction were collected to the positive and negative clusters. Finally, we estimated the size of the positive $N^{+}$ and negative $N^{-}$ clusters as the number of EEG sensors in them. The values $N^{+}$ and $N^{-}$ were compared between the conditions via a repeated-measures ANOVA.

## 3. Results

Contrasting the stimulus-related α-band power to the prestimulus baseline, we observed negative clusters in all subjects. In contrast, testing the β-band power, we found positive clusters. To analyze the spatio-temporal features of the α- and β- band power on the sensor level, we introduced seven EEG sensor-regions (see Figure 1D) and three time-intervals ($\tau_{1,2,3}$). Based on recent work [34], we assumed that $\tau_{1}=0.3$ s introduces the sensory-processing stage, whereas based on our previous work [4] $\tau_{3}=0.3$ s may reflect the decision-making stage. Moreover, we also supposed that a 0.3-s interval $\tau_{2}$ following the sensory processing stage is related to the sensory information preprocessing and the extraction of the percept-relevant stimulus features from the raw information. These intervals are schematically shown in Figure 4A,B. For all subjects $\tau_{1}$ included 0.3 s post-stimulus onset, $\tau_{2}$ expanded for 0.3–0.6 s post-stimulus onset, and $\tau_{3}$ was 0.3-s preceding RT.

### 3.1. The α-Band Activity

Figure 4 illustrates how the size of the negative cluster $N^{-}$ (group mean) depends on time and the EEG sensor regions during the processing of LA (A) and HA(B) stimuli. Presented results show that $N^{-}$ grows in time, including the EEG sensors regions from the occipital to the frontal direction.

We tested $N^{-}$ behavior via a repeated-measures ANOVA with the Ambiguity (LA, HA), Interval ($\tau_{1}$, $\tau_{2}$, $\tau_{3}$), and Region (O, P, Cp, C, Fc, F, and Fp) as within-subject factors. As a result (see Table 1), we found an insignificant main effect of Ambiguity. The main effects of Interval and Region were significant. The interaction effects Ambiguity×Interval and Interval×Region were significant. In contrast, interaction effects Ambiguity×Region and Ambiguity×Interval×Region were insignificant. Based on the observed effects, we concluded that $N^{-}$ changed between intervals differently in the different regions, but it did not depend on the ambiguity.

The post hoc analysis with the paired samples *t*-test demonstrates that $N^{-}$ significantly grows at $\tau_{2}$ when compared with $\tau_{1}$ (df=19, t=3.982, p=0.001) and at $\tau_{3}$ when compared with $\tau_{2}$ (df=19, t=5.234, p<0.001) (Figure 5A).

The post hoc analysis revealed that $N^{-}$ in the region P significantly exceeds ones for the other regions: O (df=19,t=3.4,p=0.003); C (df=19,t=4.4,p<0.001); Fc (df=19,t=5.08,p<0.001); F (df=19,t=4.9,p<0.001); Fp (df=19,t=4.5,p<0.001) excepting the area CP (df=19,t=2.03,p=0.056). The $N^{-}$, in the region CP is significantly higher than in C (df=19,t=4.3,p<0.001), FC (df=19,t=4.6,p<0.001), F (df=19,t=4.1,p=0.001) and Fp (df=19,t=3.6,p<0.001), but insignificantly differs from the O (df=19,t=1.6,p<0.116) (Figure 5B). Thus, we conclude that the α-band power decreased mainly in the occipital and parietal areas compared to the frontal regions.

Due to the significant interaction effect Ambiguity×Interval, we analyzed the $N^{-}$ difference between LA and HA stimuli separately for various intervals (Figure 5C). We found that $N^{-}$ renained similar for HA and LA stimuli at $\tau_{1}$ (df=19,t=0.1,p=0.894) and $\tau_{2}$ (df=19,t=0.3,p=0.734), whereas for $\tau_{3}$ the $N^{-}$ was significantly higher for HA stimuli when compared to LA stimuli (df=19,t=2.6,p=0.015).

Due to the significant interaction effect, Interval×Region, we compared $N^{-}$ between the different regions separately for the different time intervals (Figure 5D). We found that $N^{-}$ changed insignificantly between $\tau_{1}$ and $\tau_{2}$ for the regions F (df=19,t=1.725,p=0.101) and Fp (df=19,t=1.938,p=0.068). For the other regions, $N^{-}$ significantly increased at $\tau_{2}$ when compared to $\tau_{1}$ and at $\tau_{3}$ when compared with $\tau_{2}$.

We summarized that $N^{-}$ significantly grows during the transitions $\tau_{1}\to \tau_{2}$ and $\tau_{2}\to \tau_{3}$. This increase is observed for all EEG sensor regions excepting the regions *F* and Fp. For them $N^{-}$ changes insignificantly during the transitions $\tau_{1}\to \tau_{2}$. At the $\tau_{1,2}$, $N^{-}$ remains similar for LA and HA stimuli, whereas for $\tau_{3}$ t$N^{-}$ for HA is higher than for LA stimuli. Finally, the shape of the $N^{-}$ distribution over the EEG sensor regions remained the same for all intervals and ambiguity. It exhibited a maximum at the occipital and the parietal areas (Figure 5E).

### 3.2. The β-Band Activity

Figure 6 illustrates how the size of the positive cluster $N^{+}$ (group mean) depends on time and the EEG sensor regions during the processing of LA (A) and HA(B) stimuli. Similarly to $N^{-}$, we tested $N^{+}$ via a repeated-measures ANOVA with the Ambiguity (LA, HA), Interval ($\tau_{1}$, $\tau_{2}$, and $\tau_{3}$) and Region (O, P, Cp, C, Fc, F, and Fp) as within-subject factors.

Table 2 illustrates results of $N^{+}$ analysis. We found insignificant effects of Ambiguity and Region, and a significant main effect of Interval. The interaction effects Ambiguity×Interval and Ambiguity×Region were insignificant. Finally, we reported significant interaction effects Interval×Ambiguity and Interval×Ambiguity×Region. Based on the interaction effect Ambiguity×Interval×Region we concluded that $N^{+}$ changed differently in different regions depending on the ambiguity. Thus, we analyzed the change of $N^{+}$ between the intervals and regions separately for HA and LA stimuli.

The post hoc analysis with the paired samples *t*-test demonstrates that $N^{+}$ significantly grows at $\tau_{2}$ when compared with $\tau_{1}$ (df=19, t=2.384, p=0.028) and at $\tau_{3}$ when compared with $\tau_{2}$ (df=19, t=3.278, p=0.004) (Figure 7A).

Due to the significant interaction effect, Interval×Region, we compared $N^{+}$ between the different regions separately for the different time intervals (Figure 7B). We obtained the following results: In the region O, $N^{+}$ grows significantly between $\tau_{1}$ and $\tau_{2}$ (df=19, t=2.8, p=0.009), and between $\tau_{2}$ and $\tau_{3}$ (df=19, t=3.8, p=0.001); in the region P, $N^{+}$ also grows significantly between $\tau_{1}$ and $\tau_{2}$ (df=19, t=2.7, p=0.013), and between $\tau_{2}$ and $\tau_{3}$ (df=19, t=4.3, p<0.001); in the region CP, $N^{+}$ grows significantly between $\tau_{1}$ and $\tau_{2}$ (df=19, t=2.6, p=0.014), and between $\tau_{2}$ and $\tau_{3}$ (df=19, t=3.7, p=0.002); in the region C, $N^{+}$ remains similar for $\tau_{1}$ and $\tau_{2}$ (df=19, t=1.6, p=0.126), but grows significantly from $\tau_{2}$ to $\tau_{3}$ (df=19, t=2.9, p=0.002); in the region FC, $N^{+}$ remains similar for $\tau_{1}$ and $\tau_{2}$ (df=19, t=1.07, p=0.298), and between $\tau_{2}$ and $\tau_{3}$ (df=19, t=1.7, p=0.099); in the region F, $N^{+}$ also remains similar for $\tau_{1}$ and $\tau_{2}$ (df=19, t=1.39, p=0.178), and between $\tau_{2}$ and $\tau_{3}$ (df=19, t=0.5, p=0.585); finally, in the region Fp, $N^{+}$ remains similar for $\tau_{1}$ and $\tau_{2}$ (df=19, t=0.8, p=0.415), and between $\tau_{2}$ and $\tau_{3}$ (df=19, t=1.6, p=0.112).

Finally, due to the significant interaction effect, we compared $N^{+}$ between LA and HA stimuli for all regions at different intervals (Figure 7C). At $\tau_{1}$, *t*-test reveals no difference between HA and LA for any regions (all p>0.05). The corresponding distribution of $N^{+}$ across the skull is shown in (Figure 7D). At $\tau_{2}$, HA stimuli induces higher $N^{+}$ in the region of Fp (df=19,t=2.1,p=0.047). The corresponding distribution of $N^{+}$ across the skull for HA stimuli displays increased activity in the frontal cortex (Figure 7D). At $\tau_{3}$, HA stimuli induces higher $N^{+}$ in the region of *P* (df=19,t=4.08,p=0.001) and CP (df=19,t=2.5,p=0.019). The distributions of the $N^{+}$ across the skull reflects activation of the occipital, parietal, and temporal regions (Figure 7D).

## 4. Discussion

We analyzed EEG sensor-level wavelet power in the α and β-bands during processing ambiguous stimuli. We observed that the α-band power decreased after the stimulus onset over most sensors. In the frontal regions F and Fp, the α-band power fell before the behavioral response (interval $\tau_{3}$ in Figure 5D). Moreover, at $\tau_{3}$, it becomes more pronounced for HA stimuli than for LA stimuli (Figure 5C). In contrast, the β-band power increased after the stimulus onset. We found that HA stimuli induced high frontal β-band power at $\tau_{2}$ and high parietal β-band power at $\tau_{3}$ when compared to LA stimuli (Figure 7C).

Some studies combined fMRI and EEG analysis and reported that the parietal and frontal area’s activation accompanied the decreased α-band EEG power reflecting attentional control (see [36] for a literature review). Our results demonstrated that α-band power varied between HA and LA stimuli only at the latter processing stage, $\tau_{3}$. Thus, we suppose that HA stimuli require more attentional control to make a decision. Our results showed that frontal α-band power decreased only at $\tau_{3}$, supporting its potential role in the decision-making process [37].

The β-band activity in the frontal and parietal regions during the visual stimulus processing is associated with attentional control. Stimuli processing relies on the top-down, and bottom-up attentional mechanisms originating in the frontal and parietal cortices [38]. The bottom-up mechanism reflects involuntary attention induced by the stimulus onset (the stimulus presentation automatically makes the subject draw attention to it). The top-down mechanism shifts attention based on the knowledge about the current task. Research in monkeys [38] and humans [39] reported that neuronal activity in parietal areas at an earlier stage reflects bottom-up attention, whereases the neural activity in the prefrontal area reflects top-down attention.

In our paradigm, the stimulus onset may initiate bottom-up attention. Moreover, the morphology of LA and HA stimuli is almost the same, and we hypothesize that the bottom-up processes barely depend on ambiguity. In contrast, we suggest that the top-down mechanisms mainly contribute to goal-directed stimulus processing. On the one hand, the subject needs to draw attention to the particular properties (edges) of the Necker cube to define its interpretation. The latter requires selective attention, which according to the physiological studies, is supported by the β- band neuronal activity [40]. Selective attention is a top-down mechanism that characterizes the ability to focus cognitive resources on information relevant to our goals [21]. According to [41], enhanced neuronal activity in the frontal area serves as a neurophysiological marker of selective visual attention. On the other hand, when ambiguity is high, subjects increase their reliance on the top-down processes, such as expectations and memory, rather than on stimulus features [25]. Finally, influences of the top-down processes accompany the enhanced β-band activity [6,42].

According to our results, HA stimuli induces high frontal β-band power for 0.3–0.6 s post-stimulus onset. Ref. [34] evidenced that the sensory-processing stage might last for 0.2–0.4 s depending on the strength of the evidence. These two intervals overlap, allowing us to suppose that the frontal beta-band power controls the evidence accumulation process. Our previous work on ambiguous processing reported increasing β-band power for 0.35–0.42 s post-stimulus onset. Together with [43], we supposed that it might reflect the decision-making process. The current experimental design barely allows one to distinct the evidence accumulation and decision-making stages. At the same time, we suppose that frontal β-band power reflects top-down mechanisms that control these both stages. We speculate that ambiguous stimuli may induce the frontal β-band power evidencing internal top-down processes that control evidence accumulation to disambiguate stimulus, interpret it, and make an appropriate decision.

## 5. Conclusions

We analyzed EEG signals on the sensor level during a perceptual decision-making task. Enhancing the sensory information ambiguity, we observed a high β-band power on the frontal sensors. The literature review suggested that it might be a biomarker of top-down processes, such as attention and memory. Our study used the mixed within-trial and within-subject design. First, we found significant percept-related changes in each subject and then tested their significance at the group level. Therefore, observed β-band biomarkers are pronounced in single EEG trials and may serve as control commands for BCI. Having summarized, we highlight that increased relience on top-down processes may be an indicator of high cognitive demands, where the probability of perceptual errors increases. Thus, we suggest using these biomarkers in BCI to test the human ability to deal with perceptual uncertainties during perceptual decision-making tasks. 

## Figures and Tables

**Figure 1 sensors-21-02461-f001:**
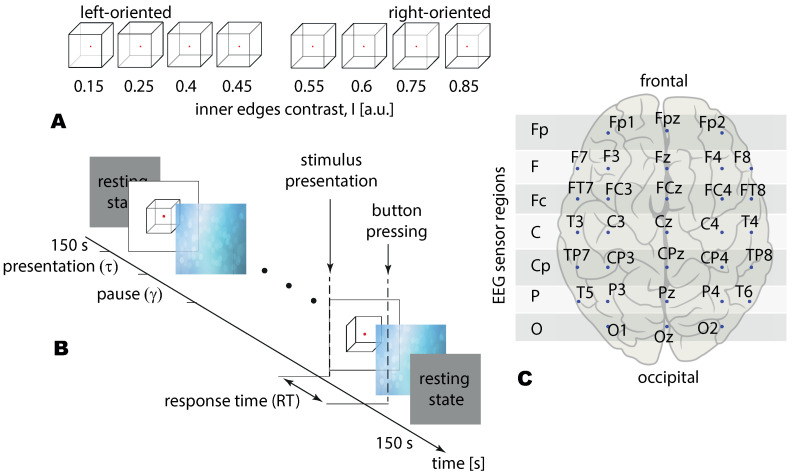
(**A**) The presented visual stimuli (Necker cubes) with the different edges contrast *I*. (**B**) Experimental design: τ∈[1,1.5] s is the time of stimulus demonstration, γ∈[3,5] s is the interval between two stimuli, and response time (RT) is the interval between presentation and response. (**C**) The 10–10 EEG (electroencephalogram) sensors layout with 31 electrodes divided into 7 EEG sensor regions are named: O, P, CP, C, Fc. F, and Fp.

**Figure 2 sensors-21-02461-f002:**
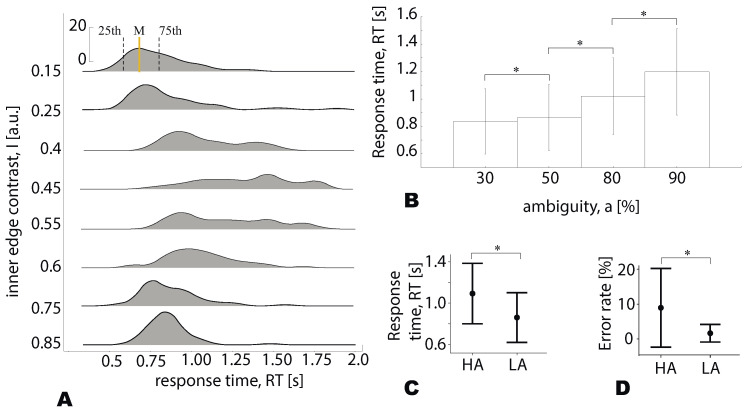
(**A**) Typical RT distributions (median, 25th, and 75th percentiles) for the different values of the inner edges contrast (on the example of a single subject). (**B**) The subject’s median RT (group mean ± SE) for the different stimulus ambiguity (* p<0.05, *t*-test with Bonferroni correction). (**C**) The subject’s median RT (group mean ± SE) for LA (low ambiguity) and HA (high ambiguity) stimuli (* p<0.001, Wilcoxon test). (**D**) The subject’s error rate (group mean ± SE) for LA and HA stimuli (* p<0.001, Wilcoxon test).

**Figure 3 sensors-21-02461-f003:**
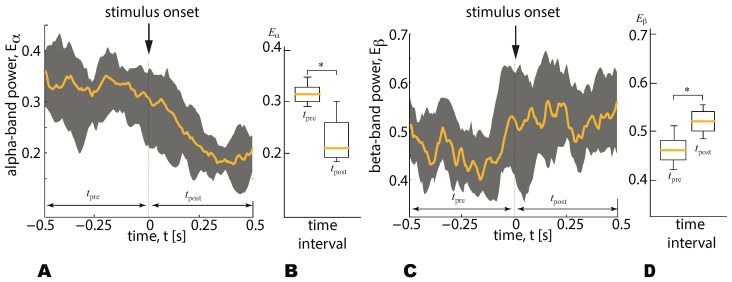
(**A**) The α-band wavelet power (mean ± SE) on the 1-s time interval associated with the stimulus presentation (dashed line shows the stimulus onset). (**B**) The α-band wavelet power averaged over the time intervals preceded ($t_{pre})$ and followed ($t_{post})$ the stimulus onset (medians, 25–75 percentiles (box) and outlines (whiskers) are shown). (**C**) The β-band wavelet power (mean ± SE) on the 1-s time interval associated with the stimulus presentation (dashed line shows the stimulus onset). (**D**) The β-band wavelet power averaged over the time intervals ($t_{pre})$ ($t_{post})$ is shown by medians (bars), 25–75 percentiles (box) and outlines (whiskers). * p<0.05 via a paired samples *t*-test.

**Figure 4 sensors-21-02461-f004:**
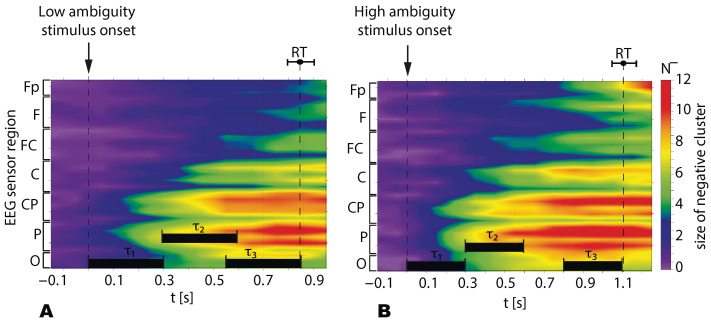
The size of the negative α-band cluster ($N^{-})$ in the different EEG sensor regions for the LA (**A**) and HA stimuli (**B**). Data are shown as the group mean, vertical dashed lines reflect the stimulus onset and the median RT, and the error bar corresponds to the standard error. The horizontal bars show the time intervals ($\tau_{1,2,3}$) for which the $N^{-}$ value is analyzed in detail.

**Figure 5 sensors-21-02461-f005:**
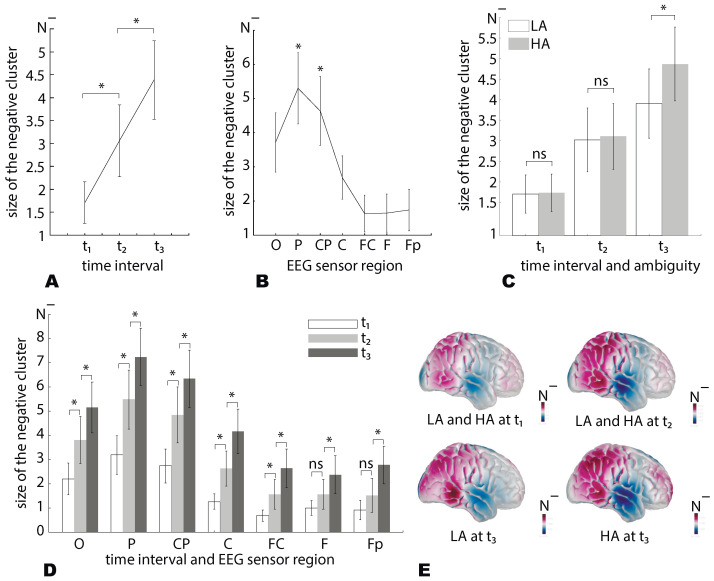
The size of negative α-band cluster, $N^{-}$ (group mean ± SE): (**A**) Averaged across all sensors and ambiguity; (**B**) averaged across the ambiguity and intervals; (**C**) averaged across the sensors; (**D**) averaged across the image ambiguity; and (**E**) distributed across the scull (head model is computed in the brainstorm [35]) * p<0.05 via a repeated measures ANOVA and the post hoc analysis with the paired-samples *t*-test.

**Figure 6 sensors-21-02461-f006:**
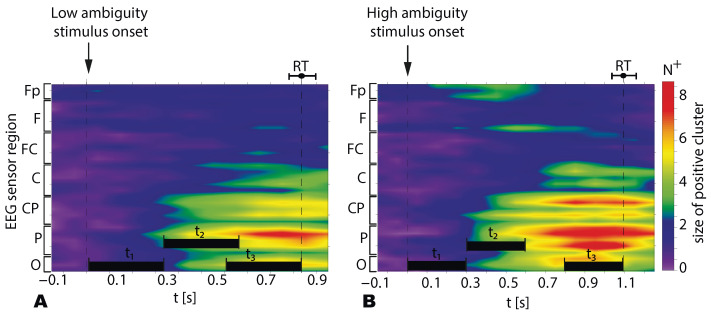
The size of the positive β-band cluster $N^{+}$ in the different EEG sensor regions for the LA (**A**) and HA stimuli (**B**). Data are shown as the group mean, vertical dashed lines reflect the stimulus onset, the median RT, and the error bars correspond to the standard error. The horizontal bars show the time intervals (0.3 s) for which the $N^{+}$ value is compared for the different ambiguities.

**Figure 7 sensors-21-02461-f007:**
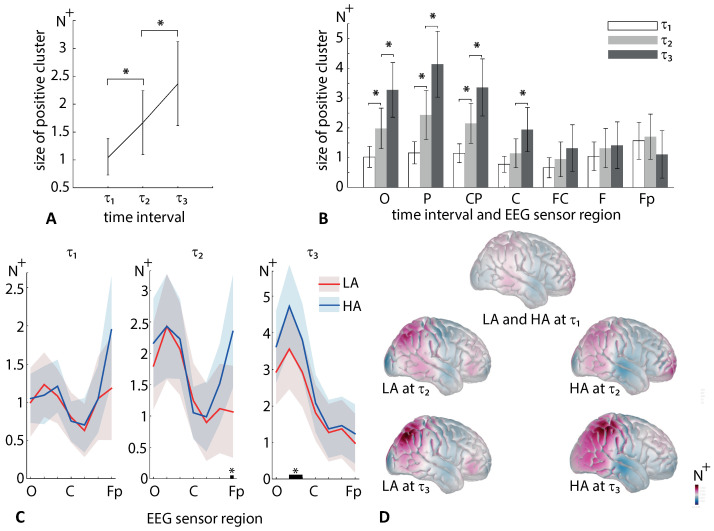
The size of the β-band positive cluster $N^{+}$ (group mean ± SE) averaged across EEG sensor regions and ambiguities (**A**); averaged across ambiguities (**B**); compared between LA and HA stimuli at three intervals (**C**); distributed across the brain surface (**D**). * p<0.05 via a repeated measures ANOVA and the post hoc analysis with the paired-samples *t*-test.

**Table 1 sensors-21-02461-t001:** Size of the negative cluster, $N^{-}$ (ANOVA summary).

Cases	dF1	dF2	F	*p*
Ambiguity (LA vs HA)	1	19	2.1	0.155
Interval ($\tau_{1}$, $\tau_{2}$, $\tau_{3}$)	1.4	26.9	27.9	0.001
Region (O, P, Cp, C, Fc, F, Fp)	2.06	39.1	15.2	0.001
Ambiguity × Interval	1.2	22.9	8.4	0.006
Interval × Region	3.7	71.7	5.7	0.001
Ambiguity × Region	4.1	78.5	0.6	0.634
Ambiguity × Interval × Region	3.9	75.8	1.6	0.168

**Table 2 sensors-21-02461-t002:** Size of the positive cluster, $N^{+}$ (ANOVA summary).

Cases	dF1	dF2	F	*p*
Ambiguity (LA vs HA)	1	19	2.5	0.129
Interval ($\tau_{1}$, $\tau_{2}$, $\tau_{3}$)	1.08	20.6	8.3	0.008
Region (O, P, Cp, C, Fc, F, Fp)	1.4	28,20	2.7	0.092
Ambiguity × Interval	1.4	28.2	3.1	0.69
Interval × Region	2.38	45.2	11.07	0.001
Ambiguity × Region	2.3	45.5	1.1	0.32
Ambiguity × Interval × Region	4.4	84.7	4.4	0.002

## Data Availability

The datasets presented in this study can be found at dx.doi.org/10.6084/m9.figshare.12292637.v2.

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
