# Peer review of "Sensor-Level Wavelet Analysis Reveals EEG Biomarkers of Perceptual Decision-Making"

_sensors, 2021, doi:10.3390/s21072461_

Round 1

Reviewer 1 Report

Overview: This study examined EEG correlates of perceptual decision making, including their time-course, frequency range, and scalp distribution. The authors found evidence for alpha- and beta-band power modulation following stimulus onset, and these patterns differed as a function of stimulus ambiguity.

Overall Assessment: I found the paper interesting and of potential value for BCI applications. The motivations were clear, and analyses were straight-forward. However, I have several concerns that preclude my recommendation for acceptance, detailed below.

  1. The introduction was missing reference to several highly relevant papers that have examined EEG during perceptual decision making, including papers examining trial-by-trial and single-trial analyses. Because the manuscript omits all of this work, it is very difficult to assess the relevance and timeliness of the presented research. Here are a few papers to start with:

Philiastides, M. G., & Sajda, P. (2007). EEG-informed fMRI reveals spatiotemporal characteristics of perceptual decision making. Journal of Neuroscience27(48), 13082-13091.

Ratcliff, R., Philiastides, M. G., & Sajda, P. (2009). Quality of evidence for perceptual decision making is indexed by trial-to-trial variability of the EEG. Proceedings of the National Academy of Sciences106(16), 6539-6544.

Nunez, M. D., Vandekerckhove, J., & Srinivasan, R. (2017). How attention influences perceptual decision making: Single-trial EEG correlates of drift-diffusion model parameters. Journal of mathematical psychology76, 117-130.

Brunyé, T. T., & Gardony, A. L. (2017). Eye tracking measures of uncertainty during perceptual decision making. International Journal of Psychophysiology120, 60-68.

Herding, J., Ludwig, S., von Lautz, A., Spitzer, B., & Blankenburg, F. (2019). Centro-parietal EEG potentials index subjective evidence and confidence during perceptual decision making. NeuroImage201, 116011.

Philiastides, M. G., & Sajda, P. (2006). Temporal characterization of the neural correlates of perceptual decision making in the human brain. Cerebral cortex16(4), 509-518.

  1. It was also challenging to relate the introduction to prior literature due to the use of the Necker Cube. Because this is a perceptual illusion, there is no objectively correct answer to each trial. This is unlike many perceptual decision making tasks, which use stimuli that have an objectively correct answer to compare performance against.
  2. Related to point #2, the authors need to describe how the ambiguity index was validated: was a pilot validation study conducted to ensure that these 8 conditions elicited the intended levels of ambiguity? Also related to the ambiguity index, it is very confusing: each polar end of the continuum is associated with low ambiguity, but the center with high ambiguity. Maybe the index should range from -1 to +1, with 0 being total ambiguity? Or, maybe the term "ambiguity" is not appropriate for the index.
  3. It was surprising to see no hypotheses laid out in the introduction. Instead, the authors reveal the results of the studies from lines 50-60 on Page 2. Unless this is a formatting policy of Sensors, this section should motivate, reveal, and justify hypotheses that will be tested in the analyses.
  4. What is the benefit of using 4 stimuli with varying levels of ambiguity in each condition (LA, HA) if you are going to collapse across them for a two-condition analysis? It would be more interesting to know whether your model can distinguish between low, moderate, and high ambiguity. As noted in the above-cited Brunye et al (2017) paper, very different perceptual and cognitive processes may underlie decision making at low, medium, and high uncertainty levels.
  5. The discussion suggests that EEG patterns are associated with bottom-up and top-down processes, and further suggests these might be associated with uncertainty and errors. However, there was no compelling way to assess errors in this study – there is no correct or incorrect answer to an illusion.

Author Response

Overview: This study examined EEG correlates of perceptual decision making, including their time-course, frequency range, and scalp distribution. The authors found evidence for alpha- and beta-band power modulation following stimulus onset, and these patterns differed as a function of stimulus ambiguity.

Overall Assessment: I found the paper interesting and of potential value for BCI applications. The motivations were clear, and analyses were straight-forward. However, I have several concerns that preclude my recommendation for acceptance, detailed below.

Q1: The introduction was missing reference to several highly relevant papers that have examined EEG during perceptual decision making, including papers examining trial-by-trial and single-trial analyses. Because the manuscript omits all of this work, it is very difficult to assess the relevance and timeliness of the presented research. Here are a few papers to start with:

Philiastides, M. G., & Sajda, P. (2007). EEG-informed fMRI reveals spatiotemporal characteristics of perceptual decision making. Journal of Neuroscience27(48), 13082-13091.

Ratcliff, R., Philiastides, M. G., & Sajda, P. (2009). Quality of evidence for perceptual decision making is indexed by trial-to-trial variability of the EEG. Proceedings of the National Academy of Sciences106(16), 6539-6544.

Nunez, M. D., Vandekerckhove, J., & Srinivasan, R. (2017). How attention influences perceptual decision making: Single-trial EEG correlates of drift-diffusion model parameters. Journal of mathematical psychology76, 117-130.

Brunyé, T. T., & Gardony, A. L. (2017). Eye tracking measures of uncertainty during perceptual decision making. International Journal of Psychophysiology120, 60-68.

Herding, J., Ludwig, S., von Lautz, A., Spitzer, B., & Blankenburg, F. (2019). Centro-parietal EEG potentials index subjective evidence and confidence during perceptual decision making. NeuroImage201, 116011.

Philiastides, M. G., & Sajda, P. (2006). Temporal characterization of the neural correlates of perceptual decision making in the human brain. Cerebral cortex16(4), 509-518.

A1: According to the comment of the Referee, we have added the corresponding discussion in the Introduction. All the changes in the revised manuscript marked by the red color.  

In their early works, Philiastides and colleagues used ERP to characterize neuronal mechanisms of perceptual decision-making. They identified ERP whose latency depended on the strength of evidence. Thus, they reported that the evidence accumulation process starts at ate early processing stages, and its duration varies as a function of the evidence strength affecting decision speed [Philiastides, 2006]. In 2007, the same authors used fMRI and reported that perceptual decision-making includes multiple spatially and temporally dissociated stages. These include early visual perception (early component), task/decision difficulty (difficulty component), and post-sensory/decision-related events (late component) [Philiastides, 2007]. In 2009, Ratcliff et al. analyzed single trials during a perceptual decision-making task. They found that the late but not the early ERP component correlated with the single-trial behavioral performance. The authors concluded that the information used to drive the decision process - that is, reflected in the late EEG component - is not the same as that which produces the early EEG response. Thus, they distinguished between neural responses to early perceptual encoding and, later, to post-sensory processing that provides the decision-relevant evidence [Ratcliff, 2009]. Note that the authors reported trial-to-trial variability in the decision performance even for identical images. It may reflect the reliance of the evidence accumulation on the human internal factors that varied through experiment. One possible mechanism modulating the human state and affecting perceptual decisions is attention. With this in mind, Nunes and colleagues evaluated attention in single trials. Their results suggested that fluctuations in attention to visual stimuli account for some of the trial-to-trial variability in the brain’s speed of evidence accumulation on each trial [Nunes, 2017]. Recent work by Herding et al. tested the effect of another human factor on the evidence accumulation process. They reported that subjective assessment of the evidence plays a crucial role in the accumulation process. According to their results, sensory input appeared to be first modulated by the subjectively perceived features and later by its absolute value (i.e., the absolute strength of evidence) [Herding, 2019]. Together, these works suggest that effects of attention and subjectiveness enhance when the evidence quality is poor, e.g., noisy or ambiguous. When the sensory information is not sufficient to reliably disambiguate the nature of a stimulus, it requires more attentional demands and efforts to acquire decision-related features. At the behavioral level, the ambiguity of sensory information influences many oculomotor variables, including fixation frequency and duration, the frequency, peak velocity, and amplitude of saccades, and phasic pupil diameter [Brunyé, 2017]. At the neural activity level, increasing ambiguity induces high $\beta-band$ power across the fronto-parietal areas, reflecting attentional modulation and the involvement of the internal processes e.g., memory and subjective experience [Herding, 2019, Engel, 2010].

Q2: It was also challenging to relate the introduction to prior literature due to the use of the Necker Cube. Because this is a perceptual illusion, there is no objectively correct answer to each trial. This is unlike many perceptual decision-making tasks, which use stimuli that have an objectively correct answer to compare performance against.

A2: Many studies used ambiguous visual stimuli, including Necker Cube to analyze spontaneous perceptual reversals. They involve the presentation of totally ambiguous stimulus whose interpretation spontaneously alternates under endogenous or exogenous factors (Kornmeier and Bach, 2006; Yokota et al., 2014). Our paradigm excluded completely ambiguous stimuli. Manipulating with the internal edge contrast, we varying ambiguity in the range where each stimulus had a particular interpretation defined by its morphology. The subjects were able to report a correct stimulus interpretation above chance. Their mean error rate was 8.95% for HA stimuli and 1.65% for LA stimuli.

  • Kornmeier, J., and Bach, M. (2006). Bistable perception—along the processing chain from ambiguous visual input to a stable percept. Int. J. Psychophysiol. 62, 345–349.
  • Yokota, Y., Minami, T., Naruse, Y., and Nakauchi, S. (2014). Neural processes in pseudo perceptual rivalry: an ERP and time-frequency approach. Neuroscience 271, 35–44.

We have added corresponding discussion in Sect. 2.5 of the manuscript

Q3: Related to point #2, the authors need to describe how the ambiguity index was validated: was a pilot validation study conducted to ensure that these 8 conditions elicited the intended levels of ambiguity? Also related to the ambiguity index, it is very confusing: each polar end of the continuum is associated with low ambiguity, but the center with high ambiguity. Maybe the index should range from -1 to +1, with 0 being total ambiguity? Or, maybe the term "ambiguity" is not appropriate for the index.

A3: We thank the referee for this remark. Indeed, parameter a reflects internal edges contrast and defines both orientation and ambiguity. Thus, we changed this definition in Sect. 2.2. First, we introduced the parameter I = {0.15,0.25,0.4,0.45,0.55,0.6,0.75,0.85} reflecting “inner edges contrast”. Then we introduced stimulus ambiguity, a in the following way. We supposed that for I=0 stimulus is unambiguously left-oriented, while for I=0.5 its features barely reflect the orientation. Varying a in the range 0-0.5, we increase the ambiguity of the left-oriented cube making it totally ambiguous at I=0.5. Thus setting a=0% ambiguity for I=0 and a=100% ambiguity for I=0.5, we suggest that stimuli with I = {0.15, 0.25, 0.4, 0.45} have a={30%, 50%, 80%, 90%} ambiguity. Similarly, for the right-oriented stimuli, we obtain that cubes with I = {0.85, 0.75, 0.6, 0.55} also have a={30%, 50%, 80%, 90%} ambiguity. Finally, to exclude effects of the stimulus orientation (including the effects associated with the formation of the motor response), we combined left- and right-oriented stimuli for each ambiguity. Fig.2,B in the revised manuscript shows that RT increases with increasing stimulus ambiguity (*p<0.05 via t-test with Bonferroni correction).

Q4: It was surprising to see no hypotheses laid out in the introduction. Instead, the authors reveal the results of the studies from lines 50-60 on Page 2. Unless this is a formatting policy of Sensors, this section should motivate, reveal, and justify hypotheses that will be tested in the analyses.

A4: A bulk of literature suggests that attention modulates neuronal activity in the alpha- and beta-frequency bands (see [Maksimenko V.A., 2019] for the literature review). On one hand, activity in these bands reflects the degree of attention in general regardless of the task [Baumgarten, 2014]. Thus, measuring the prestimulus alpha- and beta-band power enables predicting the performance of the ongoing stimuli processing. Different studies report that low alpha- and high beta-band power reflect high attention; therefore, predicting high performance and vice-versa [Gola, 2013]. On the other hand, some components of attention influence perceptual decisions when the sensory information is presented. Thus, stimulus onset involuntary draws our bottom-up attention. When the stimulus is ambiguous, its processing mobilizes top-down attentional components e.g., selective attention to facilitate the evidence accumulation process [Gazzaley, 2012]. Based on these studies, we suppose that stimulus-related attentional components also modulate the alpha- and beta- band activity on the interval following stimulus onset depending on the attentional demands, e.g., the strength of evidence. Following the Ref. [Nunes, 2017], we suppose that measuring power in these bands enables evaluating attentional demands on the single trials providing potential for BCI applications.

  • Maksimenko V.A., Frolov N.S., Hramov A.E., Runnova A.E., Grubov V.V., Kurths J., Pisarchik A.N. Neural Interactions in a Spatially-Distributed Cortical Network During Perceptual Decision-Making. Front. Behav. Neurosci.. 13, 220 (2019)
  • Baumgarten, T. J., Schnitzler, A., and Lange, J. (2014). Prestimulus alpha power influences tactile temporal perceptual discrimination and confidence in decisions. Cereb. Cortex 26, 891–903
  • Gola, M., Magnuski, M., Szumska, I., and Wróbel, A. (2013). EEG beta band activity is related to attention and attentional deficits in the visual performance of elderly subjects. Int. J. Psychophysiol. 89, 334–341.
  • Gazzaley, A.; Nobre, A.C. Top-down modulation: bridging selective attention and working memory. Trends in cognitive sciences 2012, 16, 129–135.

We have added corresponding comments in the Introduction

Q5: What is the benefit of using 4 stimuli with varying levels of ambiguity in each condition (LA, HA) if you are going to collapse across them for a two-condition analysis? It would be more interesting to know whether your model can distinguish between low, moderate, and high ambiguity. As noted in the above-cited Brunye et al (2017) paper, very different perceptual and cognitive processes may underlie decision making at low, medium, and high uncertainty levels.

A5: In this work, we further reduce the number of experimental conditions considering a={30%, 50%} as the low-ambiguity (LA) stimuli and a={80%, 90%} as high ambiguity (HA) stimuli. This simplification is based on our previous works on the Necker Cube images [Maksimenko, 2019, 2020]. It enables revealing effects of ambiguity and provides a sufficient number of trials to minimize additional effects of orientation, a bias of the presentation moment, the effect of the previously presented stimulus [Maksimenko, 2021]. At the same time, according to Brunye et al, very different perceptual and cognitive processes may underlie decision making at low, medium, and high uncertainty levels [Brunyé, 2017]. Thus, our future studies will test this effect in Necker cubes with low, moderate, and high ambiguity.

  • Maksimenko V., Kuc A., Frolov N., Kurkin S., Hramov A. Effect of repetition on the behavioral and neuronal responses to ambiguous Necker cube. Scientific Reports. 11, (2021) 3454
  • Maksimenko V.A., Kuc A., Frolov N.S., Khramova M.V., Pisarchik A.N., Hramov A.E. Dissociating Cognitive Processes During Ambiguous Information Processing in Perceptual Decision-Making. Frontiers in Behavioral Neuroscience. 14, 95 (2020)
  • Maksimenko V.A., Frolov N.S., Hramov A.E., Runnova A.E., Grubov V.V., Kurths J., Pisarchik A.N. Neural Interactions in a Spatially-Distributed Cortical Network During Perceptual Decision-Making. Front. Behav. Neurosci.. 13, 220 (2019)

Q6: The discussion suggests that EEG patterns are associated with bottom-up and top-down processes, and further suggests these might be associated with uncertainty and errors. However, there was no compelling way to assess errors in this study – there is no correct or incorrect answer to an illusion.

A6: Unlike other studies on ambiguous perception, we did not present totally ambiguous stimuli. Thus, we suggested that edges contrast reflect the orientation of the Necker cube. We instructed subjects to define the stimulus orientation and detected an error when the pressed button (left or right) did not coincide with the stimulus orientation. As a result, the subjects were able to report a correct stimulus interpretation above chance. Their mean error rate was 8.95% for HA stimuli and 1.65% for LA stimuli (See Fig.2, D in the revised manuscript).

Reviewer 2 Report

The authors of the submitted manuscript presented an approach of the analysis of time-frequency representations of EEG signals for perceptual decision-making using wavelet analysis. The authors prepared a solid background which introduces a reader to the investigated topic with sufficient support from cited literature. However, the state-of-the-art survey need to be extended according to the comments below. Then, the detailed description on the object of observation and the way of performing the experiment as well as technical details on acquisition of EEG signals together with the analyzed estimates and the processing procedure were provided. In section 3, the authors provided critical analysis of processing results using the assumed methodology with detailed comments on the connections of the ambiguity, considered here as the measure of perception, with the detected fluctuations in the EEG signals. The obtained results are summarized in the Discussion section. The manuscript is well-organized and is easy to follow. Some questions, however, need to be answered and appropriate extensions to the present version of the manuscript need to be done.

1) The topic of an analysis of alpha- and beta-bands is analyzed in many studies described in the available literature. It is worth to prepare a state-of-the-art survey on the recent studies in this topic to underline the timeliness and the originality of the current study.

2) It is better to use “transform” instead of “transformation” in the context of the sentence in lines 185-186, since “transformation” is connected with the action, while “transform” is the mathematical operation. Please check the manuscript for similar cases.

3) Please provide a justification of a selection of Morlet wavelet in the performed studies. It is well-known that the selected wavelet significantly influencing on the results during wavelet analysis, thus, such a justification is recommended to add with a focus of the specific properties of the wavelet.

4) Please also provide comments on the assumed frequency bands, especially on their connection with observed phenomena in the EEG signals.

5) It is suggested to introduce subsections in section 3.

6) Some minor language corrections are necessary before publication.

Author Response

The authors of the submitted manuscript presented an approach of the analysis of time-frequency representations of EEG signals for perceptual decision-making using wavelet analysis. The authors prepared a solid background which introduces a reader to the investigated topic with sufficient support from cited literature. However, the state-of-the-art survey need to be extended according to the comments below. Then, the detailed description on the object of observation and the way of performing the experiment as well as technical details on acquisition of EEG signals together with the analyzed estimates and the processing procedure were provided. In section 3, the authors provided critical analysis of processing results using the assumed methodology with detailed comments on the connections of the ambiguity, considered here as the measure of perception, with the detected fluctuations in the EEG signals. The obtained results are summarized in the Discussion section. The manuscript is well-organized and is easy to follow. Some questions, however, need to be answered and appropriate extensions to the present version of the manuscript need to be done.

Q1: The topic of an analysis of alpha- and beta-bands is analyzed in many studies described in the available literature. It is worth to prepare a state-of-the-art survey on the recent studies in this topic to underline the timeliness and the originality of the current study.

A1: We discussed the studies that analyze attentional modulation of the alpha- and beta-band power in the introduction.

A bulk of literature suggests that attention modulates neuronal activity in the $\alpha$- and $\beta$-frequency bands (see [Maksimenko V.A., 2019] for the literature review). On one hand, activity in these bands reflects the degree of attention in general regardless of the task [Baumgarten, 2014]. Thus, measuring the prestimulus $\alpha$- and $\beta$-band power enables predicting the performance of the ongoing stimuli processing. Different studies report that low $\alpha$- and high $\beta$-band power reflect high attention; therefore, predicting high performance and vice-versa [Gola, 2013]. On the other hand, some components of attention influence perceptual decisions when the sensory information is presented. Thus, stimulus onset involuntary draws our bottom-up attention. When the stimulus is ambiguous, its processing mobilizes top-down attentional components e.g., selective attention to facilitate the evidence accumulation process [Gazzaley, 2012]. Based on these studies, we suppose that stimulus-related attentional components also modulate the $\alpha$- and $\beta$- band activity on the interval following stimulus onset depending on the attentional demands, e.g., the strength of evidence. Following the Ref. [Nunes, 2017], we suppose that measuring power in these bands enables evaluating attentional demands on the single trials providing potential for BCI applications.

  • Maksimenko V.A., Frolov N.S., Hramov A.E., Runnova A.E., Grubov V.V., Kurths J., Pisarchik A.N. Neural Interactions in a Spatially-Distributed Cortical Network During Perceptual Decision-Making. Front. Behav. Neurosci.. 13, 220 (2019)
  • Baumgarten, T. J., Schnitzler, A., and Lange, J. (2014). Prestimulus alpha power influences tactile temporal perceptual discrimination and confidence in decisions. Cereb. Cortex 26, 891–903
  • Gola, M., Magnuski, M., Szumska, I., and Wróbel, A. (2013). EEG beta band activity is related to attention and attentional deficits in the visual performance of elderly subjects. Int. J. Psychophysiol. 89, 334–341.
  • Gazzaley, A.; Nobre, A.C. Top-down modulation: bridging selective attention and working memory. Trends in cognitive sciences 2012, 16, 129–135.

Q2: It is better to use “transform” instead of “transformation” in the context of the sentence in lines 185-186, since “transformation” is connected with the action, while “transform” is the mathematical operation. Please check the manuscript for similar cases.

A2: Following the Reviewer’s recommendation, we changed “transformation” to “transform” through the manuscript.

Q3: Please provide a justification of a selection of Morlet wavelet in the performed studies. It is well-known that the selected wavelet significantly influencing on the results during wavelet analysis, thus, such a justification is recommended to add with a focus of the specific properties of the wavelet.

A3: According to Ref. [Ogden, 2012], the Morlet wavelet has advantages in the analysis of EEG recordings that includes many rhythms and oscillatory patterns. The comparison of EEG analysis studies with different mother wavelets [Hramov, 2015] shows that the Morlet wavelet provides a clear wavelet surface and better overall resolution in the time-frequency domain.

  • Ogden, T.: Essential Wavelets for Statistical Applications and Data Analysis. Springer, New York (2012)
  • Hramov, A.E., Koronovskii, A.A., Makarov, V.A., Pavlov, A.N., Sitnikova, E.: Wavelets in Neuroscience. Springer, Berlin (2015)

Q4: Please also provide comments on the assumed frequency bands, especially on their connection with observed phenomena in the EEG signals.

A4: In line with the response to Reviewer1 and the question1 of the second Reviewer, we describe a connection between power in the alpha- and beta-bands and attention:

A bulk of literature suggests that attention modulates neuronal activity in the alpha- and beta-frequency bands (see [Maksimenko V.A., 2019] for the literature review). On one hand, activity in these bands reflects the degree of attention in general regardless of the task [Baumgarten, 2014]. Thus, measuring the prestimulus alpha- and beta-band power enables predicting the performance of the ongoing stimuli processing. Different studies report that low alpha- and high beta-band power reflect high attention; therefore, predicting high performance and vice-versa [Gola, 2013]. On the other hand, some components of attention influence perceptual decisions when the sensory information is presented. Thus, stimulus onset involuntary draws our bottom-up attention. When the stimulus is ambiguous, its processing mobilizes top-down attentional components e.g., selective attention to facilitate the evidence accumulation process [Gazzaley, 2012]. Based on these studies, we suppose that stimulus-related attentional components also modulate the alpha- and beta- band activity on the interval following stimulus onset depending on the attentional demands, e.g., the strength of evidence. Following the Ref. [Nunes, 2017], we suppose that measuring power in these bands enables evaluating attentional demands on the single trials providing potential for BCI applications.

Q5: It is suggested to introduce subsections in section 3.

A5: We introduced subsections in the section 3, as the Reviewer suggests.

Q6: Some minor language corrections are necessary before publication.

A6: Following this remark, we check spelling through the text.

Round 2

Reviewer 1 Report

The authors have done a comprehensive job responding to my concerns and revising their manuscript. I believe it is in a much more suitable position for publication at this point, and I appreciate the authors' effort. The only suggestion is for minor text editing for typos and grammar.